# Identifying the Trends of Urinary microRNAs within Extracellular Vesicles for Esophageal Cancer

**DOI:** 10.3390/cancers16091698

**Published:** 2024-04-27

**Authors:** Kazuhiko Hisaoka, Satoru Matsuda, Kodai Minoura, Hiroki Yamaguchi, Yuki Ichikawa, Mika Mizunuma, Ryota Kobayashi, Yosuke Morimoto, Masashi Takeuchi, Kazumasa Fukuda, Rieko Nakamura, Shutaro Hori, Taigi Yamazaki, Takehiko Sambe, Hirofumi Kawakubo, Yuko Kitagawa

**Affiliations:** 1Department of Surgery, Keio University School of Medicine, 35 Shinanomachi, Shinjuku-ku, Tokyo 160-8582, Japan; 2Craif Inc., Tokyo 113-0034, Japan; 3Department of Clinical Research and Development, Graduate School of Pharmacy, Showa University, Tokyo 142-8555, Japan; 4Department of Clinical Pharmacology, Graduate School of Medicine, Showa University, Tokyo 142-8555, Japan

**Keywords:** urinary microRNAs, esophageal squamous cell carcinoma, liquid biopsy, biomarkers

## Abstract

**Simple Summary:**

The advancement of multidisciplinary treatment has increased the need to develop a test to monitor tumor burden during treatment. We herein analyzed urinary microRNAs within extracellular vesicles from patients with esophageal squamous cell carcinoma (ESCC) and normal individuals using a microarray, and identified 18 microRNAs which were significantly expressed in ESCC patients.

**Abstract:**

**Background**: The advancement of multidisciplinary treatment has increased the need to develop tests to monitor tumor burden during treatment. We herein analyzed urinary microRNAs within extracellular vesicles from patients with esophageal squamous cell carcinoma (ESCC) and normal individuals using a microarray. **Methods:** Patients with advanced ESCC who underwent esophagectomy (A), endoscopic submucosal resection (ESD) (B), and healthy donors (C) were included. Based on microRNA expression among the groups (Analysis 1), microRNAs with significant differences between groups A and C were selected (Analysis 2). Of these candidates, microRNAs in which the change between A and C was consistent with the change between B and C were selected for downstream analysis (Analysis 3). Finally, microRNA expression was validated in patients with recurrence from A (exploratory analysis). **Results:** For analysis 1, 205 microRNAs were selected. For Analyses 2 and 3, the changes in 18 microRNAs were consistent with changes in tumor burden as determined by clinical imaging and pathological findings. The AUC for the detection of ESCC using 18 microRNAs was 0.72. In exploratory analysis, three of eighteen microRNAs exhibited a concordant trend with recurrence. **Conclusions:** The current study identified the urinary microRNAs which were significantly expressed in ESCC patients. Validation study is warranted to evaluate whether these microRNAs could reflect tumor burden during multidisciplinary treatment for ESCC.

## 1. Introduction

Esophageal cancer is a dismal disease, since it metastasizes even in the early stage [1,2,3], resulting in the sixth leading cause of cancer death [4]. According to a nationwide survey by the Japan Esophageal Society (8019 cases treated in 2013 and analyzed in 2019) [5], the incidence of esophageal cancer in Japan shows a male-to-female ratio of approximately 5.4:1, with males being more affected. The onset age is typically between the ages of 60 and 70, accounting for around 70% of all cases. 

The outcomes for esophageal cancer patients have improved with the development of multidisciplinary treatments [6,7,8]. Currently, the standard treatment for resectable esophageal cancer primarily includes surgery, endoscopic treatment, chemotherapy, immunotherapy, and radiotherapy [6,9,10]. For surgically resectable advanced esophageal cancer, based on the CROSS trial, neoadjuvant chemoradiotherapy (NACRT) followed by surgery has become a standard worldwide [8,11]. On the other hand, in Japan, neoadjuvant chemotherapy (NAC) followed by surgery was shown to be beneficial compared to adjuvant following surgery (JCOG9907) [12]. More recently, it was shown that the intensification of NAC with triplet chemotherapy further improved overall survival comparing to doublet in JCOG1109 [13,14]. After curative treatment, aggressive follow-up is required to detect recurrent disease and determine an indication for further treatment. Currently, computed tomography (CT) and esophagogastroduodenoscopy (EGD) are used for surveillance during treatment, but these modalities are invasive, labor-intensive, costly, and involve radiation. Furthermore, they are not capable of detecting minimal residual diseases (MRDs). Carcinoembryonic antigen and squamous cell carcinoma-associated antigen are used as tumor markers for diagnosis and follow-up; however, the positive rates are only 13.1% and 24.5%, respectively, and the specificity is not high [15].

In recent years, liquid biopsy has attracted attention as a tumor monitoring method [16]. During a surgically resectable stage, evaluating MRD is important for selecting the indication for surgery after preoperative chemotherapy and the need for postoperative chemotherapy. Liquid biopsy is expected to become a reliable detection modality to guide this process [17,18]. Of the various modalities for liquid biopsy, microRNA has been suggested as a potential biomarker for the diagnosis and prediction of cancer [19,20]. Although approximately 2000 types of microRNAs have been discovered in humans, only 200–300 urinary microRNAs can be extracted by conventional methods [21]. Therefore, although the expression levels of serum and plasma microRNAs in esophageal cancer patients may be useful biomarkers for the diagnosis of esophageal cancer, there are few urinary microRNAs.

As an original analytical method for microRNAs in urinary extracellular vesicles (EVs), an EV collection device that combines ZnO nanowire scaffolds and microfluidic channels was developed [22]. We have successfully extracted approximately 1500 types of urinary microRNAs using this device. Compared with blood samples, urine samples have the advantage of being noninvasive and easy to handle. In order to utilize urinary microRNA to monitor tumor burden during multidisciplinary treatment, as an initial step, it would be valuable if we could identify microRNAs specifically expressed in esophageal cancer. In this study, to identify urinary microRNAs specific for ESCC, we evaluated the microRNA levels in urine from patients with ESCC and healthy donors. Collecting healthy donors with heavy smoking history could play an important role in improving the specificity to esophageal cancer. For exploratory analysis, changes in urinary microRNAs were longitudinally assessed before and after curative treatment and at recurrence.

## 2. Materials and Methods

### 2.1. Patient Selection and Treatment

To identify urinary microRNAs that reflect tumor burden, patients with ESCC and normal individuals were included. We constructed three cohorts, and all patients were enrolled prospectively. Because this was an exploratory study, we set the total number of patients in advance at 10 for superficial esophageal cancer (SEC), 20 for advanced esophageal cancer (AEC), and 20 for healthy donors. Patients with ESCC were classified into AEC (Cohort A) and SEC (Cohort B). Normal individuals were defined as Cohort C, which was divided into two groups: C-1, those who planned to undergo inguinal hernia repair, and C-2, those who were healthy without a history of smoking.

The inclusion requirements for Cohort A were as follows: (1) histologically diagnosed as ESCC; (2) plan to undergo curative surgery or chemoradiotherapy at Keio University Hospital between May 2020 and July 2021; and (3) stage II, III, or IV including cT4b. The inclusion requirements of Cohort B were as follows: (1) histologically diagnosed as ESCC; (2) plan to undergo endoscopic submucosal resection between May 2021 and July 2021; and (3) cStage 0. The inclusion criteria of Cohort C-1 were as follows: (1) healthy individuals; (2) plan to undergo inguinal hernia repair between June 2021 and August 2021. The inclusion criteria of Cohort C-2 were as follows: (1) healthy individuals; (2) received health examination at Showa University Medical Checkup Center in February 2021; and (3) Brinkman index of 500 or higher.

Staging was evaluated using the Union for International Cancer Control TNM staging system 8th edition. The study was approved by the Institutional Review Board of Keio University School of Medicine (IRB no. 20190057). Written informed consent was obtained from all patients before enrollment.

### 2.2. Treatment Progress and Follow-Up

Based on the JCOG9907 study, the preoperative treatment was cisplatin plus 5-FU (CF) therapy twice every three weeks for resectable advanced ESCC. Alternatively, a regimen of docetaxel and cisplatin plus 5-FU (DCF) was provided every three weeks for three cycles [23,24]. After neoadjuvant chemotherapy (NAC), standard surgical procedure involved trans-thoracic esophagectomy with either right thoracotomy or thoracoscopy, followed by gastric tube reconstruction through the posterior mediastinal route. Mediastinal lymph nodes, including those adjacent to bilateral recurrent nerves, and abdominal lymph nodes, such as paracardial lymph nodes and those along the lesser curvature and left gastric artery, were dissected. Postoperative follow-up consisted of esophagogastroduodenoscopy (EGD) and computed tomography (CT) every six months for a duration of five years post-surgery.

### 2.3. Sample Collection

Urine samples were collected from AEC and SEC patients two weeks before treatment and frozen at −80 degrees C within 1–2 h after collection. In two patients who experienced postoperative recurrences, samples were collected before NAC, postoperatively, and at recurrence to evaluate longitudinal urinary microRNA monitoring.

### 2.4. Isolating microRNAs from Urine Utilizing the Nanowire Device

We isolated microRNAs from a 1 mL urine sample using a nanowire-microfluidic de-vice, following centrifugation (15 min, 4 °C, 3000× *g*) to pellet cells and cellular debris. Components utilized included an fCOP resin microfluidic substrate, COP resin substrate, two stainless steel holders, and PEEK tubes. Post-device assembly, the inlet PEEK tube was linked to a syringe pump (KDS-200, KD Scientific Inc., Holliston, MA, USA) for urine and lysis buffer (Cell Lysis Buffer M, Wako Pure Chemical Industries Ltd., Hiroshima-shi, Japan) introduction. Simultaneously, the outlet PEEK tube was placed into an RNase-free microfuge tube (Ep-pendorf AG., Hamburg, Germany) for microRNA-containing solution and urine flow-through collection. The microRNA-containing solution, extracted with lysis buffer, underwent purification using the SeraMir Exosome RNA Purification Column Kit (System Biosciences Inc., Palo Alto, CA, USA) as per the manufacturer’s instructions. Nanowire devises were obtained by Craif Inc. and urine samples were processed by them.

### 2.5. Performing Microarray Analysis to Examine microRNA Expression

A comprehensive analysis of microRNA expression was conducted using the 3D Gene microRNA Labeling Kit and the 3D-Gene Human miRNA Oligo Chip (Toray Industries, Inc., Tokyo, Japan), capable of detecting 2632 microRNAs registered in miRBase release 22. Fluorescent signals for each microarray spot were captured using a 3D-Gene Microarray Scanner (Toray Industries, Inc.) and digitized using the accompanying digitizing application “Extraction” (Toray Industries, Inc.). Raw signals underwent background (BG) subtraction, where the true signal was statistically inferred based on a normal distribution assumption for the BG signal and an exponential distribution assumption for the true signal. microRNAs with a (BG-subtracted signal) >26 in at least 50% of the samples were retained for downstream analyses, while others were excluded. Quantile normalization was performed following BG subtraction and removal of low-expression microRNAs, followed by log2 transformation.

### 2.6. Statistical Analysis

For the comparison of mean values between the two groups, the Mann–Whitney U-test was conducted to test statistical significance. A *p*-value < 5% was considered statistically significant unless otherwise mentioned. The log2 fold-change value was calculated as the difference of the group-wise mean of the expression values between the two groups. Cohen’s d value was used for the effect size between the two groups.

For binary classification, a logistic regression model with an L2 penalty was fitted with the microRNA expression profiles as input variables and the presence of cancer as the target variable. The performance of the model was evaluated by leave-one-out cross-validation.

All statistical and machine learning analyses of the microRNA expression profiles were performed using the BG-subtracted, normalized, and log2 transformed values of the fluorescent signal as described in “Microarray analysis of microRNA expression”.

AUC analysis used a logistic regression model with normalized expression levels of 18 microRNAs to discriminate between esophageal cancer patients and healthy donors. The prediction scores were obtained via leave-one-out cross-validation, and the area under the receiver operating characteristic curve was calculated with the R package ROCR.

## 3. Results

### 3.1. Patient Characteristics and the Urinary microRNA Profile

Patient characteristics and clinicopathological factors are listed in Table 1. To identify candidate urinary microRNA biomarkers, we compared changes in urinary microRNA expression between AEC and HC, SEC and HC, and AEC and SEC. microRNA expression changes were assessed using fold-change (Figure 1), with 130 microRNAs showing an increase and 75 showing a decrease (Figure 2). The microRNAs that exhibited statistically significant changes were pared down by comparing AEC vs. HC with a *p*-value of <5% and a Cohen’s d > 0.5. As a result, 57 microRNAs showed a significant increase in expression, and 11 exhibited a decrease. Finally, from these 68 microRNAs, we identified microRNAs that reflected the tumor burden of esophageal cancer in the SEC vs. HC comparison. Fifteen microRNAs showed a significant increase in expression between the two groups (Figure 3), whereas three exhibited a decrease (Figure 3). Thus, we identified 18 microRNAs that may reflect tumor burden during multidisciplinary treatment for ESCC (Appendix A). The area under the curve for the diagnosis of healthy donors and esophageal cancer patients (SEC and AEC) was 0.72 (Table 2).

### 3.2. Longitudinal Evaluation of Urinary microRNA Levels

We analyzed urine samples longitudinally collected from two patients with postoperative recurrence at three time points: before NAC, after surgery, and at recurrence. Case 1 was a 61-year-old woman with cT4bN2M0 cStage IVa who underwent thoracoscopic esophagectomy after three courses of DCF therapy (Appendix A). A pathological examination indicated pT3N2M0 pStage III. Postoperative recurrence was observed in the supraclavicular lymph nodes and lymph nodes along the gastric conduit seven months after surgery. The following 17 of 18 microRNAs exhibited fluctuations reflecting the clinical course: miR.197.3p, miR.3085.3p, miR.371b.3p, miR.3940.3p, miR.4323, miR.4665.3p, miR.4763.5p, miR.6751.3p, miR.6775.3p, miR.6785.3p, miR.6800.3p, miR.6824.3p, miR.6848.3p, miR.6872.3p, and miR.939.3p, and they were decreased in expression postoperatively and increased at relapse, whereas miR.6831.5p and miR.10401.5p increased postoperatively and decreased at recurrence (Appendix A). Surgery was performed after three courses of neoadjuvant chemotherapy. Postoperative recurrence was observed in supraclavicular lymph nodes and lymph nodes along the gastric conduit seven months after surgery.

Case 2 involved a 69-year-old man with cT3N4 M0 cStage IVa who underwent thoracoscopic esophagectomy after three courses of DCF therapy (Appendix A). A pathological examination indicated pT3N3M0 at stage III. Recurrence in the mediastinal lymph nodes was detected two months following surgery. Four of the eighteen identified microRNAs exhibited fluctuations reflecting the clinical course. MiR.371b.3p, miR.4323, and miR.6824.3p were decreased after surgery, whereas miR.6831.5p and miR.6877.5p were increased postoperatively and decreased at recurrence (Appendix A). Surgery was performed after three courses of neoadjuvant chemotherapy. Mediastinal lymph nodes were detected two months after surgery.

Three out of eighteen microRNAs showed consistent trends longitudinally in both patients. hsa-miR-4323 and hsa-miR-6824-3p were decreased in expression postoperatively and increased at recurrence, whereas hsa-miR-6831-5p was increased in expression postoperatively and decreased at recurrence.

## 4. Discussion

We identified 18 microRNAs which were significantly expressed in esophageal cancer patients compared to a healthy cohort. The area under the curve for the diagnosis of 20 healthy donors and 30 esophageal cancer patients (endoscopically adapted esophageal cancer + AEC) was 0.72. These results indicated that there were urinary miNAs which were specific for ESCC patients. Furthermore, the expression for each microRNA reflected disease progression, since magnitudes became larger as diseases advanced. When evaluated in the exploratory analysis, microRNA changes before NAC, after surgery, and at the time of recurrence were determined in two patients with postoperative recurrence. Three of the eighteen microRNAs exhibited a similar trend of change in the early postoperative period and at the time of recurrence. The microRNAs that showed consistent trends longitudinally in both patients were hsa-miR-4323, hsa-miR-6824-3p, and hsa-miR-6831-5p. These microRNAs have been reported to be associated with various types of cancer. The hsa-miR-4323 was significantly decreased in colorectal cancer, suggesting its potential involvement in the progression of the cancer [25]. The hsa-miR-6831-5p has been reported to be an independent risk factor for breast cancer resistance [26]. Although validation study is mandatory, the results indicated that microRNAs in urinary EVs may serve as a tumor monitoring method for ESCC.

Tumor monitoring may be useful for meticulous follow-up following curative treatment. Currently, CT and EGD are used to detect recurrent disease after curative treatment, such as surgery and chemoradiotherapy, but their accuracy is insufficient, since they cannot detect minimal residual diseases. Furthermore, due to the non-specific enlargement of lymph nodes, the presence of radiological mass does not always indicate the existence of tumor cells [27,28]. In order to quantify the tumor burden and monitor its trends systemically, liquid biopsy using tumor-derived nucleic acids in liquid samples, such as blood, urine, and pleural fluid, has attracted attention. We previously reported that the circulating tumor DNA (ctDNA) reflected the response of NAC in esophageal cancer. Additionally, the ESCC patients who were classified as ctDNA-positive showed significantly worse prognosis than those who were ctDNA-negative, indicating that ctDNA can be one of the clinically relevant biomarkers to evaluate MRD. On the other hand, evaluating ctDNA in blood still requires blood draws and its sensitivity might not be enough when tumor volume is low. Therefore, it would be valuable if we could identify less invasive ways which can be evaluated repeatedly. Several previous reports have examined urinary microRNAs. Urabe et al. identified two microRNAs (miR-373-5p and miR-6766-5p) that are useful for the diagnosis of bladder pain syndrome/interstitial cystitis [21] and Kitano et al. established a diagnostic index based on the individualized values of twenty-three microRNAs to differentiate between patients with central nervous system tumors and normal individuals [29]. Although there have been scattered reports on the usefulness of blood-derived ctDNA and microRNA as biomarkers for esophageal cancer [30], few studies have assessed whether urinary microRNA reflects the presence of tumors.

An advantage of using urine is that specimens can be readily collected noninvasively from any location. Diagnosis and follow-up of esophageal and other gastrointestinal cancers are performed by endoscopy and CT; however, invasiveness, high cost, radiation exposure, and the time and effort required to visit a hospital are impediments. Liquid biopsy using urine samples may facilitate the meticulous follow-up required with repeatedly collected patient samples, especially after curative treatment, in which no obvious targeted lesions exist.

For the development of multidisciplinary treatment of ESCC, preoperative treatment, such as NAC or NACRT, is used worldwide [8,13]. In Japan, a randomized trial, JCOG9907, demonstrated that neoadjuvant CF significantly improves overall survival compared with adjuvant chemotherapy [12]. Subsequently, a three-arm phase III trial comparing CF with DCF and CF with CR plus radiation as preoperative treatment for locally AEC (JCOG1109) indicated that neoadjuvant DCF was significantly superior to CF for overall survival of ESCC patients [13]. Furthermore, the real-world evidence encouraged recommendations for DCF as a standard regimen for NAC-based treatment of ESCC [31]. While the indication of neoadjuvant DCF therapy should be carefully evaluated, especially in aged patients, since its survival advantage might be insufficient due to the negative impact of postoperative complications on survival, it has been recognized as the standard treatment for AEC [32]. Recently, the CheckMate-577 trial demonstrated the benefit of nivolumab as adjuvant chemotherapy following NACRT plus radical surgery [33]. Because immune checkpoint inhibitors could induce irreversible immune-related adverse events and there is a concern for cost effectiveness, an accurate tumor burden monitoring with liquid biopsy may be useful for selecting an appropriate candidate for adjuvant therapy.

This study had several limitations. First, it was exploratory and conducted in a single center with a limited number of cases. Although the number of patients for each cohort was determined without a power calculation, it was set before our sample collection began to exclude selection biases. The microRNAs extracted in this study were identified by a microarray analysis, which can qualitatively measure the expression of hundreds of thousands of genes at once; however, the downside of handling several genes simultaneously is that false positives may occur. More accurate data can be obtained by examining the expression of individual genes using RT-PCR or RNA sequencing.

## 5. Conclusions

The current study identified the urinary microRNAs which were significantly expressed in ESCC patients. Validation study is warranted to evaluate whether these microRNAs could reflect tumor burden during multidisciplinary treatment for ESCC.

## Figures and Tables

**Figure 1 cancers-16-01698-f001:**
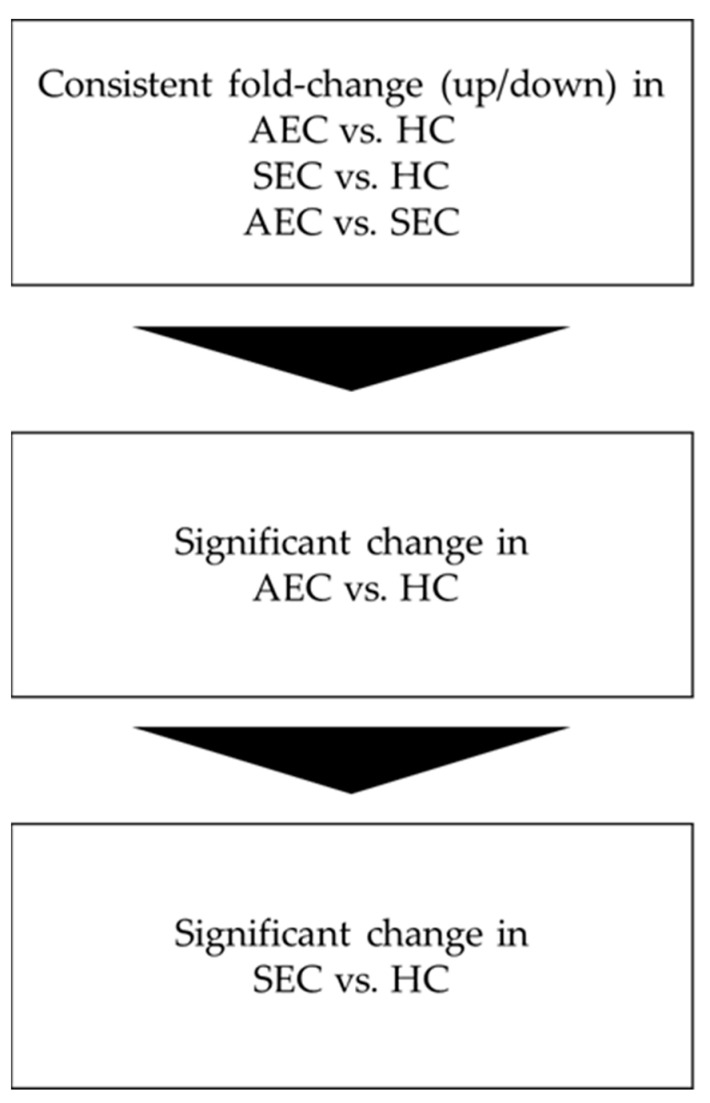
Analysis flow. AEC, advanced esophageal cancer; HC, healthy cohort; SEC, superficial esophageal cancer.

**Figure 2 cancers-16-01698-f002:**
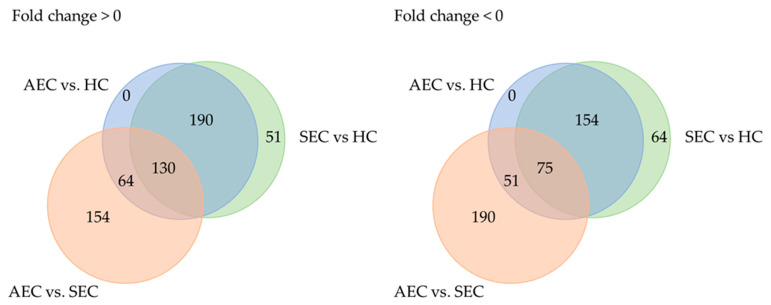
Trends in microRNA expression among groups. AEC, advanced esophageal cancer; HC, healthy cohort; SEC, superficial esophageal cancer.

**Figure 3 cancers-16-01698-f003:**
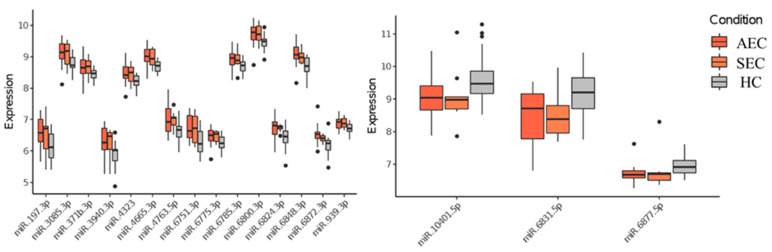
Urinary microRNAs which were consistent with changes in tumor burden. HC, healthy cohort; SEC, superficial esophageal cancer; AEC, advanced esophageal cancer.

**Table 1 cancers-16-01698-t001:** Patient characteristics.

		Healthy	AEC	SEC
*n*		20	20	10
Age	Mean (SD)	66.8 (9.4)	68.2 (7.5)	70.5 (6.4)
Gender	Male	16 (80%)	17 (85%)	10 (100%)
	Female	4 (20%)	3 (15%)	0 (0%)
Gender ratio	Male/Female	4	5.7	NA
Alcohol	Current use	16 (80%)	18 (90%)	6 (60%)
	Past use	0 (0%)	2 (10%)	4 (40%)
	Never	4 (20%)	0 (0%)	0 (0%9
Smoking history	Current use	10 (50%)	5 (25%)	3 (30%)
	Past used	5 (25%)	14 (70%)	5 (50%)
	Never	5 (25%)	1 (5%)	2 (20%)
Stage	0	0	0	10 (100%)
	I	0	1	0
	II	0	2	0
	III	0	9	0
	IVA	0	5	0
	IVB	0	3	0
	NA	20 (100%)	0	0
Recurrence		0	2 (10%)	0

AEC, advanced esophageal cancer; SEC, superficial esophageal cancer; NA, not applicable.

**Table 2 cancers-16-01698-t002:** Area under the curve analysis.

	AUC (95% C.I.)
Higher in AEC/SEC Compared to HC	
hsa-miR-197-3p	0.730 (0.592–0.868)
hsa-miR-3085-3p	0.787 (0.663–0.912)
hsa-miR-371b-3p	0.743 (0.607–0.878)
hsa-miR-3940-3p	0.746 (0.611–0.881)
hsa-miR-4323	0.737 (0.600–0.874)
hsa-miR-4665-3p	0.782 (0.656–0.908)
hsa-miR-4763-5p	0.734 (0.597–0.872)
hsa-miR-6751-3p	0.772 (0.643–0.900)
hsa-miR-6775-3p	0.770 (0.641–0.899)
hsa-miR-6785-3p	0.741 (0.605–0.877)
hsa-miR-6800-3p	0.753 (0.620–0.886)
hsa-miR-6824-3p	0.772 (0.643–0.900)
hsa-miR-6848-3p	0.795 (0.673–0.917)
hsa-miR-6872-3p	0.797 (0.675–0.919)
hsa-miR-939-3p	0.749 (0.615–0.883)
Lower in ESCC	
hsa-miR-10401-5p	0.735 (0.598–0.872)
hsa-miR-6831-5p	0.725 (0.586–0.864)
hsa-miR-6877-5p	0.764 (0.634–0.895)

AUC, area under the curve; ESCC, esophageal squamous cell carcinoma.

## Data Availability

The data presented in this study are available in this article.

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
