# Peer review of "Identifying the Trends of Urinary microRNAs within Extracellular Vesicles for Esophageal Cancer"

_cancers, 2024, doi:10.3390/cancers16091698_

Round 1
Reviewer 1 Report
Comments and Suggestions for Authors
The authors investigated whether microRNAs within extracellular vesicles in urine from patients with early and advanced ESCC can be used as biomarkers for tumour burden as an alternative to microRNAs obtained from blood. They compared those results with the microRNAsose in urine from healthy individuals.
Although this was an exploratory study, the weak point is that only one sample was collected from ESCC and healthy patients. Assuming all ESCC-treated patients return to the hospital for post-treatment control, a second urine sample could have been collected. Those data could have strengthened the quality of the results described in the manuscript. Now, they show data at three-time points of only two patients with a recurrence.
Other comments:
- Mention in the Introduction that the prevalence of ESCC in Japan is in men (smoking, drinking) approximately 3-4x higher than in women, which would explain, at the same time, the men/woman ratio in Table 1.
- M+M section 2.3: rewrite this section. Remove the first and second sentences and start with "Urine samples were collected from AEC and SEC patients two weeks before treatment and frozen at -80 degrees C within 1–2 h after collection. In two patients ....".
- M+M Section 2.4: Was the nanowire device obtained from Craif Inc? If so, mention it.
- Legends are required for the Figures 1-3.
- Figure 1 is unnecessary.
- Figure 2 needs clarification: what is meant by early vs healthy, late vs healthy, and late vs early? Be consistent in the manuscript and call early ->SEC, late ->AEC, and healthy ->HC. Perhaps each circle could have a separate colour or pattern
- Table 2, be consistent. Do not change to ESCC when using AEC and SEC throughout the manuscript.
- What is the justification for the cutoff level of AUC of microRNAs for 0.72?
Minor issues:
- page 2, line 5: at the end of the sentence "... following surgery." add between brackets (JCOG9907 study) and reference 11 (missing) because that study comes back later in the text,
- page 3, line 6, add reference number 11 at end of sentence,
- page 8, Discussion, 2nd paragraph, lines 8-9: the reference is missing.
- page 9, Conclusions, line 2: typo 2x "with ESCC".
Comments on the Quality of English Language
The American English is OK, but some editing is needed to create smoother sentences.
Reviewer 2 Report
Comments and Suggestions for Authors
In the last years, the development of non-invasive methods to monitor disease course has attracted great attention. The evaluation of MicroRNA in urine represent a promising approach, as suggested by some publications.
The aim of this study is monitoring tumor burden using urinary microRNA in esophageal cancer. MicroRNA purification from urine is based on a nanowire microfluidic device fabricated by the authors and previously published (ref. 21).
Starting from a series of 30 esophageal cancer patients and 20 healthy donors, authors identified 18 miRNA for potential monitoring of the disease.
Subsequently, the 18 selected miRNA were evaluated to monitor tumor burden in 2 patients.
In these 2 patients, 3 of 18 miRNA displayed a similar trend of variation during treatment and at recurrence. Authors conclude that urinary microRNA extracted with a nanowire device reflect the tumor burden of esophageal squamous cell carcinomas and may serve as tumor monitoring method.
Comment:
The major problem of this manuscript is that only 2 patients were monitored. This number is too small to draw conclusions.
Other comments:
In the abstract cohort A is defined as advanced esophageal cancer and cohort B is superficial esophageal cancer, but in material and methods the definition is the opposite.
In Figure 2 (VENN Diagram) it is difficult to understand which group is represented by each circle. Probably a color image could help in distinguishing the groups.
In Figure 3 it should be specified the category represented by each boxplot (eg. Pink is… Gray is…..)
There are some typographical errors (e.g SCE in place of SEC, AUC in place of AEC, Has-miR in place of Hsa-miR, …)
Comments on the Quality of English LanguageThere are some typographical errors (e.g SCE in place of SEC, AUC in place of AEC, Has-miR in place of Hsa-miR, …)
Round 2
Reviewer 2 Report
Comments and Suggestions for Authors
The revised version has been improved.
The new version is in better alignment with the data presented in the article.
Comments on the Quality of English LanguageMinor editing required